# Total and Differential Cell Counts as a Tool to Identify Intramammary Infections in Cows after Calving

**DOI:** 10.3390/ani11030727

**Published:** 2021-03-07

**Authors:** Alfonso Zecconi, Gabriele Meroni, Valerio Sora, Roberto Mattina, Micaela Cipolla, Lucio Zanini

**Affiliations:** 1Department of Biomedical, Surgical and Dental Sciences—One Health Unit, University of Milano, Via Pascal 36, 20133 Milano, Italy; gabriele.meroni@unimi.it (G.M.); valerio.sora@unimi.it (V.S.); roberto.mattina@unimi.it (R.M.); 2Associazione Regionale Allevatori Lombardia, Via Kennedy 30, 26013 Crema, Italy; m.cipolla.cmh@gmail.com (M.C.); luciozanini@gmail.com (L.Z.)

**Keywords:** mastitis, differential cell count, total cell count, periparturient period, diagnosis

## Abstract

**Simple Summary:**

Mastitis is a costly disease and needs to be identified as soon as possible to reduce the negative effect on milk quality and quantity and to maximize the chance of cure when an antimicrobial therapy is applied. Bacteriological diagnosis is expensive and not easily available in some areas, therefore approaches to reduce the number of samples to be taken, focusing the interest on cows with higher chances to have an intramammary infections are desirable. The results of our study based on a large database of quarter milk samples analyses including bacteriological analysis, total (SCC) and differential (DSCC) cell count in the first 5–30 days after calving suggest a new and sustainable approach. Indeed, a marker (PLCC) calculated by multiplying SCC and DSCC showed to have the lowest cost when applied to identify udder quarters at risk to have an intramammary infection due to major pathogens. Moreover, this approach as well as the one based on SCC became a benefit when the prevalence of these infections exceeds 10%, and it be of high interest, when selective dry cow therapy is applied, to improve animal health at the herd level.

**Abstract:**

Milk differential somatic cells count (DSCC), made possible under field conditions by the recent availability of a high-throughput milk analyzer may represent an improvement in mastitis diagnosis. While an increasing number of studies reports data on DSCC on individual cow samples, very few concerns DSCC from quarter milk samples. This paper reports for the first time the results of a retrospective study aiming to assess the performance of total (SCC), DSCC, and a novel calculated marker (PLCC) measured on quarter milk samples as a method to identify cows at risk for intramammary infection (IMI) in the first 30 days after calving. Overall, 14,586 valid quarter milk samples (3658 cows) taken in the first 30 days of lactation were considered. Quarters with major pathogens (MP) IMI, as expected, showed significantly higher means for SCC, DSCC, and PLCC. The accuracy, sensitivity, and specificity of the diagnosis based on different cut-offs calculated by ROC analysis are relatively close among DSCC, PLCC, and SCC (up to cut-off of 200,000 cells/mL). However, decision-tree analysis which includes the costs of analysis, but also the costs of the actions taken after test results showed as PLCC has the lowest cost among the three markers, and PLCC and SCC are cost effective when MP prevalence is higher than 6–10%. This diagnostic approach is of high interest particularly when selective dry cow therapy is applied to improve animal health at the herd level.

## 1. Introduction

Mastitis is the inflammation of one or more quarters of the mammary gland, almost always caused by bacteria [1]. This definition implies that the diagnosis of mastitis should be based on either bacteriologic analysis or on the increase of inflammatory markers, or both. The inflammatory marker predominantly applied worldwide is the measurement of somatic cells (SCC) either by direct or indirect methods [2,3]. Moreover, SCC is generally cheaper than microbiological analysis and may be used to select cow/quarters to be bacteriologically analyzed, even if this approach may be inaccurate [4,5]. This latter approach is becoming even more challenging since the average SCC in dairy herds is generally decreasing, while the proportion of cows with low SCC is increasing [6]. Indeed, low SCC doesn’t assure that cow/quarter are not infected [6,7]. Currently, a level of 200,000 cells/mL is considered the threshold to identify subclinical mastitis [7,8,9], but effects of udder inflammation were observed even below 100,000 cells/mL [6,10,11,12], and nearly half of contagious pathogens have a SCC below 100,000 cells/mL [13,14].

A potential improvement in mastitis diagnosis may be represented by milk differential somatic cells count (DSCC), and the scientific interest on DSCC for mastitis diagnosis is confirmed by the increasing number of papers on this topic observed in the last 10–15 years [15]. However, several studies performed on mastitis diagnosis with DSCC gave controversial results, very likely due to the different diagnostic methods applied, to the different milk samples selected and to the different fractions of milk considered [12,16,17,18,19]. 

The recent availability of a high-throughput milk analyzer allows to overcome these latter problems and to apply DSCC to milk samples routinely. Several studies are available, mainly on the application of DSCC on individual milk samples (Dairy Herd Improvement samples) [6,20,21,22,23], but only one on quarter milk samples [24].

Recently the application of selective dry-cow therapy (SDCT) received a lot of interest. This approach showed to be efficient, cost effective, and to be able to reduce the use of antimicrobials [25]. However, to avoid that new intramammary infections mainly, but not exclusively, in untreated cows will develop in clinical mastitis and/or in the spread of contagious pathogens, the control of udder health post-calving is recommended in order to improve animal health at the herd level [26]. The microbiological analysis of all the cows post-calving will reduce the economic benefit of SDCT, and methods to identify cow at risk in this phase are needed to decrease the number of cows sampled. Studies on SCC and CMT as a tool to select cows showed to have a low accuracy [4]. In our knowledge, data on the use of DSCC from quarter milk samples to select cow at risk after calving are not available.

To contribute to fill this gap, this paper reports the results of a retrospective study aiming to assess the use of SCC, DSCC, and a novel marker (PLCC) obtained by multiplying SCC and DSCC, as a method to identify cows at risk for intramammary infection (IMI) in the first 5-30 days after calving.

## 2. Materials and Methods 

### 2.1. Database

A database including the results of bacteriological and cytological analyses in >30,000 quarter milk samples from 18 Lombardy dairy herds was considered for this study. Herds were participating to herd health management programs or to field studies on the application of SDCT. All the cows in the participating herds were sampled routinely at quarter level at least once. A subset including only samples from cows within 5–30 days in milk (DIM) was selected from the database.

### 2.2. Microbiological Analyses

Quarter milk samples (QMS) were taken following the procedure described by N.M.C., 2017 [27], delivered refrigerated to Regional Breeders Association (ARAL) laboratories where bacteriological analyses were performed by streaking 0.01 mL of QMS on blood agar plate with 5% (*v*/*v*) bovine blood according to N.M.C., 2017 [27]. After incubation (18–24 h at 37 °C) the colonies recovered were identified by routine biochemical and immunological methods [28]. All the samples were taken at least 5 days after calving to minimize the influence of colostrum, as suggested [9,29].

Based on the results of the bacteriological analysis a quarter was classified as follows:(a)Positive for an IMI due to Major contagious pathogens (*Str.agalactiae*, *S.aureus*) when 1 or more colonies were isolated.(b)Positive for an IMI due to Major environmental pathogens (*Str.uberis*, *Str.dysgalactiae*) when 5 or more colonies of the same species were isolated.(c)Positive for an IMI due to Gram negative pathogens (*E. coli*, *Klebsiella* spp., other coliforms) when 5 or more colonies of the same species were isolated.(d)Positive for an IMI due to Minor pathogens (Coagulase negative *Staphylococcus species* other environmental *Streptococcus species*, *Enterococcus species*): 10 or more colonies of the same genus were isolated.(e)Negative when no colonies were recovered, or their number was below the values defined in the previous classes.(f)Contaminated when ≥3 different colony types were isolated from the milk sample.

The previous classification is based on pathogenic characteristics of the bacteria, and, mainly, on an approach, currently applied in our laboratories, aiming to increase the judiciously application of antimicrobials, restricting their use in cow infected with bacteria (included in -a- and -b- classes) with very high chances to induce a clinical mastitis and/or to spread contagious infection within the herd that can be (cost-) effectively controlled by antimicrobial treatment. 

### 2.3. Cytological Analyses

Quarter milk analyses were also analyzed to assess SCC and DSCC and were carried by the means of Fossomatic™ 7DC (Foss A/S, Hillerød, Denmark). The analysis of DSCC was based on Foss DSCC Method Cell Staining (international patent PCT/EP2010/065615-Holm, 2012), which allows to identify within a milk sample the macrophages (MAC) and the combination of PMN and LYM, as described by Damm et al. [30]. The DSCC is expressed as the combined proportion (%) of PMN and LYM on the overall count of milk cells.

### 2.4. Statistical Analysis

The data were collected in a database including: herdID, cowID, parity (PAR), days in milk (DIM), quarter (FL, FR, RR, RL), SCC, and DSCC, a new variable (PLCC) calculated by multiplying DSCC * SCC and the results of bacteriological analysis as described before.

Data were analyzed by calculation of ROC curves on Xlstat 2020.5.1 (Addinsoft. New York, NY, USA) to define the optimal cut-off values to identify cows at risk for IMI by the markers considered (SCC, DSCC, PLCC). The response variable considered (disease) was the presence of an IMI due to major pathogens (MP), Gram-negative (GN) and minor pathogens (LP). This statistical procedure allows to represent the evolution of the proportion of true positive cases (sensitivity) as a function of the proportion of false positives cases (corresponding to 1 minus specificity), and to evaluate a binary classifier test to diagnose a disease. The analysis allows to calculate for each cut-off point the following variables: Sensitivity (Se), specificity (Sp), positive predictive value (PPV), negative predictive value (NPV), true positive (TP), true negative (TN), false positive (FP), false negative (FN), and accuracy. 

In the case of SCC, in addition to the optimal cut-off values calculated, the performance of the test was also analyzed applying cut-off values of 100,000 cells/mL, 200,000 cells/mL, and 400,000 cells/mL, representing the linear score values respectively of 3, 4 and 5 [31].

To assess the payoff and probability of each possible diagnostic path based on the different markers and the related cut-off values, a decision-tree analysis was performed by the means of PrecisionTree™ software (ver. 8.1.0. 2020 Palisade Corp. Ithaca, NY, USA) The software determines the best decision to make at each decision node. Once your decision tree is complete, it creates a full decision analysis statistics report and its comparison with alternative decisions. Including the cost of each decision. The input values for this analysis were: Cytological analysis (SCC + DSCC) 2€: bacteriological analysis 6€ (lowest commercial cost in Italy); cost of a contagious pathogen IMI (related only to decrease in milk yield), 350€; cost of a mild clinical mastitis 100€. Both these latter costs were estimated in in previous surveys in Italian dairy herds [32,33].

## 3. Results

### 3.1. Data Description 

Samples with SCC/DSCC missing data and contaminated bacteriological results were withdrawn from the initial dataset obtaining 14,586 valid quarter milk samples (3658 cows) taken in the first 30 days of lactation.

The results of bacteriological analyses were summarized in Table 1. The results showed as 70.6% of the samples were bacteriologically negative. Among the positive ones, minor pathogens, and more specifically coagulase negative Staphylococci (CNS) are the most frequently isolated bacteria (22.7%), representing 77.4% of the positive samples.

Major pathogens were recovered only in 1.5% of the samples with *S. aureus* representing 2.2% of the positive samples, and major environmental Streptococci, *Str. uberis* and *Str. dysgalactiae*, recovered respectively in 2.6% and 0.3% of the positive samples.

Gram negative bacteria were isolated in 3.6% of the samples, while other minor pathogens such as other environmental Streptococci and Enterococci were isolated respectively in 1.0% and 0.2% of the samples.

When data on total and differential cell counts were considered (Table 2), bacteriologically negative samples had, as expected, the lowest mean values for SCC, DSCC, and PLCC (significant at α = 0.05). Samples harboring MP, on the other end, showed the significantly highest means for the three cytological markers considered.

Significant differences were also observed for SCC and PLCC in GN positive samples when compared with negative and LP positive quarters, however the numerical difference among these mean values were relatively small.

### 3.2. ROC Curves and Cut-Off Values for SCC, DSCC and PLCC

ROC curves were calculated to assess the capability of SCC, DSCC, and PLCC to identify quarters with IMI due to the three main categories of pathogens considered (PP, GN, and LP). The results were reported in Table 3. In the case of SCC, performance values were calculated for the optimal value defined by ROC analysis and for the three levels usually applied in practice, as described in the material and methods section, while for DSCC and PLCC the cut-off values were only the optimal ones calculated on ROC analysis.

A large variability among the cut-off values was observed for all the cellular markers in relation to the pathogen groups. As expected, also test performances largely varied in relation both to cellular markers considered and pathogens group. The highest accuracy values were observed for any of the markers considered when the tests were applied to identify major pathogens IMI, while the lowest accuracy values were observed for GN detection.

The highest accuracy was observed for major pathogens detection applying a cut-off of 400.000 cells/mL for SCC; these results is mainly to the very high level of Sp. The highest accuracies for all the pathogen groups, with very high Sp, but with very low Se (range 0.134–0.382) were also observed for this cut-off value. DSCC showed lower accuracy values than SCC at any cut-off levels, also when compared with PLCC. This latter marker had accuracy values comparable to SCC when MP an LP were considered, but accuracy was relatively lower when GN were considered. Sensitivity was relatively low for all the pathogens groups and for all the markers being in the range 0.382–0.682 for MP, 0.161–0.732 for GN and 0.134–0.444 for minor pathogens. 

### 3.3. SCC, DSCC and PLCC as Markers for Major Pathogens in Primiparous and Pluriparous Cows

Recent studies showed as the pattern of milk leucocytes is influenced by cow parity [20]. Therefore, the ROC analysis was also performed on two subsets including respectively primiparous cows (5077 samples) and pluriparous cows (9509). Based on the previous results, the diagnostic performances of the three cellular markers were assessed in identifying MP infected quarters (Table 4). As expected, the optimal cut-off values for all the three markers considered were different in the two subsets. Higher cut-off values were observed in primiparous cows when compared with pluriparous ones. 

When SCC were considered, the cut-off level of 400,000 cells/mL confirmed to have the highest accuracy among all the markers and cut-off values, but also the lowest Se. The highest Se was observed for the SCC, 0.727 (cut-off value 70,000 cells/mL) and 0.685 (cut-off value 66,000 cells/mL) respectively in primiparous and pluriparous cows. DSCC optimal cut-off value in primiparous cows has a lower Se than SCC, but Sp was 20% higher. In pluriparous cows both Se and Sp were comparable with the values calculated for SCC. PLCC showed a similar pattern as DSCC, but accuracies were generally higher than the ones observed for SCC and DSCC.

From practical point of view, the values of Se and Sp define how many samples should be taken and bacteriologically analyzed (TP + FP). In primiparous cows the lowest number was observed for DSCC and PLCC, respectively 784 (sampling fraction 15.4%) and 831 (sampling fraction 16.4%), while in the case of SCC (optimal cut-off) this number was 1560 (sampling fraction 30.7%). In pluriparous cows the lowest number of potential samples was also observed for PLCC (2168; sampling fraction 22.8%), while 3120 samples (sampling fraction 32.8%) were estimated for SCC and 3330 samples (sampling fraction 35.0%) were estimated for DSCC.

### 3.4. Preventive Costs

The selection of quarters to be bacteriologically analyzed to identify the presence of MP may be considered as a preventive measure and it can be calculated the relative cost by the means of a decision-tree analysis [34,35]. The results of these analyses were summarized in Table 5, while an example of the decision tree applied was reported in Figure 1. PLCC showed to have the lowest cost among all the markers and all the cut-off values considered. The lower cost is the result of both the relative low number of samples analyzed and of the chances to identify MP among these samples, thus preventing the effects of contagious IMI and the outcome of clinical mastitis.

The previous analyses suggested to apply different cut-off values to primiparous and pluriparous cows; therefore, preventive costs are expected to be different. Indeed, when sample selection was based on optimal cut-off value for PLCC, the lowest cost for primiparous cows was 2.55 €/cow, while it was 2.62 €/cow in pluriparous ones.

To estimate the performance of this approach in epidemiological scenario different from this study, a sensitivity analysis was performed applying the optimal values for the three markers selected, at different prevalence of MP. The analysis showed as the approach based on DSCC was always a cost (increasing as prevalence increases) (data not shown). The approaches based on SCC and PLCC became a benefit when the prevalence of MP IMI in primiparous cows is higher than 6% and 10% respectively for SCC and PLCC (Figure 2), whereas in pluriparous cows the breakeven points were respectively 6% and 8% (Figure 3). The same analysis based on the changes in the proportion of contagious and environmental pathogens (at the same prevalence of the study) did not show to results in a benefit, with relatively small changes in the cost (data not shown).

## 4. Discussion

The importance of monitoring udder health in cows after calving is well known [4], and it became even more important when SDCT is applied [26]. This latter approach may decrease significantly antimicrobial treatments, thus reducing the risk for AMR and the production costs [25]. These positive effects may be impaired by an increase of IMI and/or clinical mastitis and by the costs of analyzing all the cows after calving. Therefore, new sustainable approaches must be developed within to maintain the benefits and keep under control the costs. 

This retrospective study aimed to collect information on new approaches based on SCC and DSCC to select cows to be bacteriologically analyzed in the first 5–30 days after calving. In our knowledge, this is the first study available on this topic, and include, by far, the largest number of samples.

The data on bacteriological analysis confirmed that IMI are present in cows after freshening [36,37], and CNS are the prevalent bacteria species, while contagious pathogens and major environmental Streptococci have a lower prevalence, when compared with the other bacteria species. The low prevalence of contagious and environmental pathogens may be due to the health status of the included herds. Indeed, they are all participating to field investigations, including studies on applying selective dry cow therapy, and, consequently, they have relatively high standards of health and management. 

The most practical and sustainable approach to select cows to be bacteriologically analyzed, until recently, was to identify cows at risk by the means of direct or indirect SCC measurement [4]. This approach is feasible in many different conditions, but it has a relatively low efficiency being affected by an unbalanced Se/Sp ratio, independently of the cut-off applied [4,7]. 

More recently, a high throughput instrument able to perform DSCC became available which increases dramatically the opportunities to have DSCC data at a sustainable cost [6,30,38]. The DSCC has been applied extensively on individual cow sampling, but still rarely on quarter milk samples [24]. Moreover, our recent study [20] showed as the total amount of PMN and lymphocytes (P + L) has a pattern that is only partially related to SCC and their number is kept relatively steady in healthy cows. These observations suggest calculating a new marker (PLCC) obtained by multiplying SCC for DSCC of each sample, and describing the overall content of P + L per ml.

Quarters with MP IMI, as expected, were associated to significant higher means for the three cellular markers considered, whereas negative quarters showed significant lower means for all the markers, when compared to quarters with IMI. The low mean levels for all the three cellular markers observed in samples with IMI due to GN and LP suggest a relatively low inflammatory response and support the prudent approach of avoiding antimicrobial treatment [37,39]. 

For what concerns SCC, the results of our study confirm previous ones on the accuracy of direct and indirect SCC measured either at lab or herd level in detecting IMI [4,36,40]. Moreover, the accuracies of DSCC and PLCC were generally similar to SCC (max cut-off level 200,000 cells/mL) in primiparous cows and somewhat lower in pluriparous cows. These values are comparable to results recently reported by Schwarz et al. [24] on cows close to drying-off. The different accuracies observed between primiparous cows as well as the different cut-off levels confirm previous observations on the different cellular immune response in primiparous cows which is generally more intense when compared with older cows [20]. 

The other important factor to be analyzed to decide which diagnostic approach should be applied is to evaluate the costs or benefits attainable. This assessment can be performed by the means of decision tree analysis that is a practical way to assess the efficiency of intervention protocols under field conditions [40,41]. This analysis considers the performances of the tests, including the number of FP and FN outcome and the benefit deriving from a correct detection of an IMI, as well as the costs of unidentified cases. The results of this analysis showed as PLCC has the lowest cost among all the markers considered, independently of the cut-off selected, and in both primiparous and pluriparous cows, suggesting that the “interaction” between SCC and DSCC may give a more accurate picture on the health status of the udder than SCC and DSCC alone or in parallel [24]. When sensitivity analysis was applied to the optimal values for the three markers selected, at different prevalence of MP, the analysis showed as the approach based on SCC and PLCC became a benefit increasing the MP prevalence. These analyses showed as PLCC approach is more cost-effective than SCC when MP prevalence is low (<10%). In our knowledge, this is the first study estimating the cost/benefits of selecting cow at risk for MP IMI based on SCC and DSCC alone and in combination with SCC, therefore it is not possible to make any comparison, but it would be of great interest to furthermore investigate these aspects in mastitis diagnosis. 

## 5. Conclusions

There is an increasing need to have diagnostic approaches for mastitis feasible and sustainable. The availability of tools able to measure differential cell count efficiently and in a relatively inexpensive way allows to develop new approaches. In this study the new marker, PLCC, based on SCC multiply by DSCC showed to have accuracy, Se and Sp comparable with SCC and DSCC. Moreover, it showed to have a lower cost when included in a decision-tree analysis including the potential effects of a positive diagnosis and the negative ones when an IMI goes undetected. This diagnostic approach is of high interest when applied to cows soon after calving in herd with an elevated health status (low PM prevalence), particularly when selective dry cow therapy is applied to maintain and, possibly, improve animal health at the herd level.

## Figures and Tables

**Figure 1 animals-11-00727-f001:**
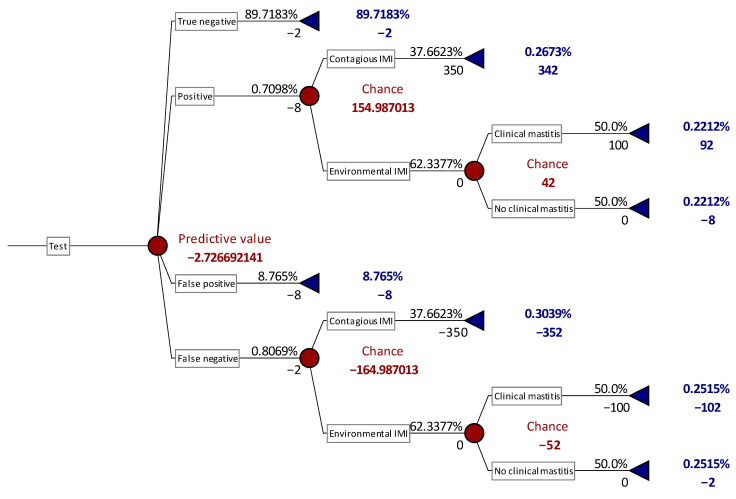
Decision tree example based on SCC cut-off of 400,000 cells/mL for major pathogens IMI detection in primiparous. Epidemiological and diagnostic performances are the same of this study and input values are reported in Material and Methods section. Numbers alone (red or blue character) represent costs (€), while numbers followed by % represent the probability of the status described in each line of the graph.

**Figure 2 animals-11-00727-f002:**
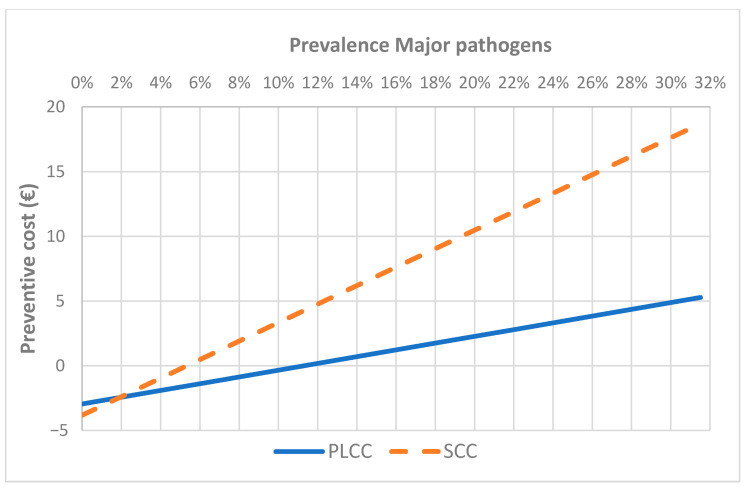
Results of sensitivity analysis applying SCC and PLCC optimal cut-off values in primiparous cows at different prevalence of major pathogens IMI.

**Figure 3 animals-11-00727-f003:**
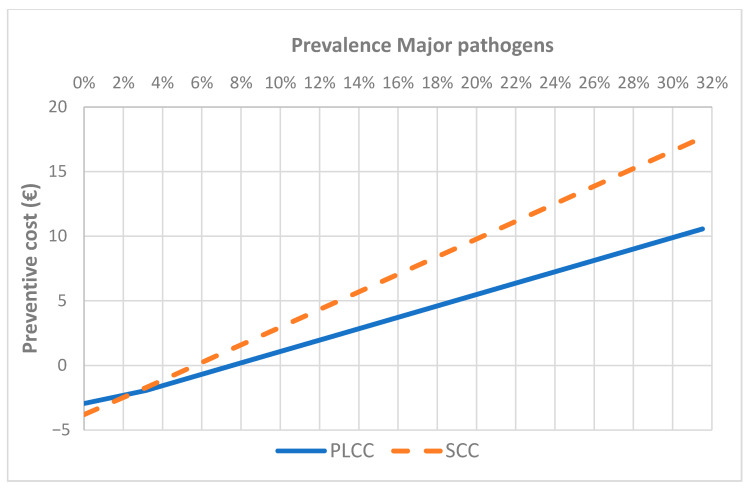
Results of sensitivity analysis applying SCC and PLCC optimal cut-off values in pluriparous cows at different prevalence of major pathogens IMI.

**Table 1 animals-11-00727-t001:** Distribution of the results of 14,586 bacteriological analyses performed on quarter milk samples of cows in the first 5–30 days of lactation.

Pathogen Isolated	Number of Quarters	% of Total Samples	% of Positive Quarters
NO GROWTH	10,302	70.6	
Major pathogens	220	1.5	5.1
*Staphylococcus aureus*	96	0.7	2.2
*Streptococcus uberis*	113	0.8	2.6
*Streptococcus dysgalactiae*	11	0.1	0.3
Gram-negative	522	3.6	12.2
*Escherichia coli*	244	1.7	5.7
Other coliforms	278	1.9	6.5
Minor pathogens	3542	24.3	82.7
Coagulase-negative Staphylococci	3314	22.7	77.4
Environmental Streptococci (Streptococco other than *Str.uberis* and *Str.dygalactiae*)	143	1.0	3.3
*Enterococcus* spp	35	0.2	0.8
Others	50	0.3	1.2
TOTAL POSITIVE	4284	29.4	

**Table 2 animals-11-00727-t002:** Distribution of the results of 14,586 cytological analyses performed on quarter milk samples of cows in the first 30 days of lactation.

Pathogen Isolated	SCS ± Std.dev. (Log10/mL)	DSCC ± Std.dev. (%)	PLCC ± Std.dev. (Log10/mL)
NO GROWTH	4.51 ± 0.70 ^a,1^	54.48 ± 18.89 ^a^	4.21 ± 0.778 ^a^
Major pathogens	5.24 ± 0.85 ^b^	65.12 ± 16.57 ^b^	5.04 ± 1.03 ^b^
*Staphylococcus aureus*	5.21 ± 0.92	65.58 ± 14.50	5.01 ± 0.98
*Streptococcus uberis*	5.27 ± 0.96	64.32 ± 18.24	5.03 ± 1.05
*Streptococcus dysgalactiae*	5.23 ± 1.30	69.27 ± 15.97	5.12 ± 1.37
Gram negative	4.80 ± 0.85 ^c^	58.40 ± 16.50 ^c^	4.55 ± 0.85 ^c^
*Escherichia coli*	4.71 ± 0.74	55.78 ± 16.87	4.41 ± 0.82
Other coliforms	4.87 ± 0.86	60.83 ± 5.84	4.63 ± 1.00
Minor pathogens	4.72 ± 0.74 ^d^	57.89 ± 17.20 ^c^	4.46 ± 0.83 ^d^
Coagulase-negative staphylococci	4.72 ± 0.74	57.93 ± 17.12	4.45 ± 0.82
Environmental Streptococci (Streptococci other than *Str. uberis* and *Str. dysgalactiae*)	4.65 ± 0.85	56.85 ± 17.18	4.37 ± 0.95
*Enterococcus* spp	4.91 ± 0.86	56.17 ± 18.68	4.61 ± 1.00
Others	4.80 ± 0.75	58.85 ± 16.51	4.55 ± 0.82

^1^ Columns with different superscripts are statistically different (*p* < 0.05). Data were analysed comparing the four major classes (MP, GN, LP and negative.

**Table 3 animals-11-00727-t003:** Summary results of ROC analyses on the application of the three cytological markers (somatic cell count, SCC; differential cell count DSCC; PMN and lymphocyte count, PLCC) on the identifications of quarter with intramammary infection (IMI) due to major pathogens, minor pathogens and Gram-negative bacteria.

Patho Gens	Marker	Level	Sensitivity	Lower Bound (95%)	Upper Bound (95%)	Specificity	Lower Bound (95%)	Upper Bound (95%)	PPV ^1^	NPV ^1^	TP ^1^	TN ^1^	FP^1^	FN ^1^	Accuracy
MAJOR PATHOGENS	SCC (n/mL)	70,000	0.682	0.617	0.740	0.704	0.697	0.712	0.034	0.993	150	10,120	4246	70	0.704
		100,000	0.605	0.539	0.667	0.765	0.758	0.772	0.038	0.992	133	10,994	3372	87	0.763
		200,000	0.514	0.448	0.579	0.848	0.842	0.854	0.049	0.991	113	12,187	2179	107	0.843
		400,000	0.382	0.320	0.448	0.901	0.896	0.906	0.056	0.990	84	12,950	1416	136	0.894
	DSCC (%)	67%	0.577	0.511	0.641	0.725	0.717	0.732	0.031	0.991	127	10,411	3955	93	0.722
	PLCC (n/mL)	65,960	0.614	0.548	0.675	0.779	0.772	0.785	0.041	0.992	135	11,188	3178	85	0.776
GRAM NEGA TIVE	SCC (n/mL)	20,000	0.741	0.702	0.777	0.391	0.383	0.399	0.043	0.976	387	5500	8564	135	0.404
		100,000	0.330	0.291	0.371	0.763	0.756	0.770	0.049	0.968	172	10,731	3333	350	0.747
		200,000	0.249	0.214	0.288	0.846	0.840	0.852	0.057	0.968	130	11,902	2162	392	0.825
		400,000	0.161	0.132	0.195	0.899	0.894	0.904	0.056	0.967	84	12,648	1416	438	0.873
	DSCC (%)	50.4%	0.722	0.682	0.759	0.384	0.376	0.392	0.042	0.974	377	5403	8661	145	0.396
	PLCC (n/mL)	13,914	0.649	0.607	0.689	0.489	0.481	0.497	0.045	0.974	339	6880	7184	183	0.495
MINOR PATHOGENS	SCC (n/mL)	53,000	0.444	0.427	0.460	0.672	0.664	0.681	0.292	0.799	1508	7523	3664	1891	0.619
		100,000	0.325	0.309	0.340	0.785	0.778	0.793	0.315	0.793	1103	8785	2402	2296	0.678
		200,000	0.211	0.198	0.225	0.859	0.853	0.866	0.313	0.782	717	9612	1575	2682	0.708
		400,000	0.134	0.123	0.146	0.907	0.901	0.912	0.304	0.775	456	10,143	1044	2943	0.727
	DSCC (%)	63,400	0.420	0.404	0.437	0.677	0.668	0.686	0.283	0.793	1428	7573	3614	1971	0.617
	PLCC (n/ mL)	32700,000	0.430	0.413	0.446	0.699	0.690	0.707	0.302	0.801	1460	7817	3370	1939	0.636

^1^ PPV: Positive predictive value, NPV: Negative predictive value, TP: True positive, TN: True negative, FP: False positive, FN: False negative.

**Table 4 animals-11-00727-t004:** Summary results of ROC analyses on the application of the three cytological markers (somatic cell count, SCC; differential cell count DSCC; PMN and lymphocyte count, PLCC) on the identifications of quarter with IMI due to major pathogens, in primiparous or pluriparous cows.

Marker	Value	Sensitivity	Lower Bound (95%)	Upper Bound (95%)	Specificity	Lower Bound (95%)	Upper Bound (95%)	PPV ^1^	NPV ^1^	TP ^1^	TN ^1^	FP ^1^	FN ^1^	Accuracy
	Primiparous cows													
SCC (n/mL)	70,000	0.727	0.618	0.814	0.699	0.686	0.712	0.036	0.994	56	3496	1504	21	0.700
	100,000	0.623	0.511	0.723	0.762	0.750	0.773	0.039	0.992	48	3809	1191	29	0.760
	200,000	0.506	0.397	0.615	0.856	0.846	0.865	0.051	0.991	39	4280	720	38	0.851
	400,000	0.468	0.360	0.578	0.911	0.903	0.919	0.075	0.991	36	4555	445	41	0.904
DSCC (%)	73.5	0.468	0.360	0.578	0.850	0.840	0.860	0.046	0.990	36	4252	748	41	0.845
PLCC (n/mL)	108,000	0.558	0.447	0.664	0.842	0.832	0.852	0.052	0.992	43	4212	788	34	0.838
	Pluriparous cows													
SCC (n/mL)	63,000	0.685	0.605	0.756	0.690	0.680	0.699	0.033	0.993	98	6462	2904	45	0.690
	100,000	0.594	0.512	0.671	0.767	0.758	0.776	0.038	0.992	85	7185	2181	58	0.765
	200,000	0.517	0.436	0.598	0.844	0.837	0.851	0.048	0.991	74	7907	1459	69	0.839
	400,000	0.336	0.264	0.417	0.896	0.890	0.902	0.047	0.989	48	8395	971	95	0.888
DSCC (%)	63.5	0.650	0.569	0.724	0.654	0.645	0.664	0.028	0.992	93	6129	3237	50	0.654
PLCC (n/mL)	66,000	0.622	0.541	0.698	0.778	0.769	0.786	0.041	0.993	89	7287	2079	54	0.776

^1^ PPV: Positive predictive value, NPV: Negative predictive value, TP: True positive, TN: True negative, FP false positive, FN: False negative.

**Table 5 animals-11-00727-t005:** Decision tree analysis results applying the different cellular markers and relative cut-off values for detection of cows at risk for major pathogens IMI.

Marker	Level	N. Test	Sampling Fraction (%)	Preventive Cost (€)	P Outcome among Test Positive		P Outcome among Test Negative	
					Contagious IMI	Clinical Mastitis	Contagious IMI	Clinical Mastitis
**Primiparous cows**								
SCC (n/mL)	63,000	1669	32.87	−2.85	0.42	0.34	0.16	0.13
	70,000	1560	30.73	−2.72	0.42	0.35	0.16	0.13
	100,000	1239	24.40	−2.75	0.35	0.29	0.21	0.15
	200,000	759	14.95	−2.86	0.29	0.24	0.28	0.23
	400,000	481	9.47	−2.72	0.27	0.22	0.30	0.25
DSCC (%)	63.5%	1693	33.35	−3.52	0.34	0.28	0.23	0.19
	73.5%	784	15.44	−4.08	0.28	0.22	0.30	0.25
PLCC (n/mL)	66,000	1145	22.55	−2.87	0.34	0.28	0.23	0.29
	108,000	831	16.37	−2.55	0.34	0.28	0.24	0.19
**Pluriparous cows**								
SCC	63,000	3002	31.57	−2.83	0.48	0.27	0.28	0.13
	70,000	2760	29.03	−2.84	0.43	0.26	0.24	0.14
	100,000	2266	23.84	−2.89	0.42	0.24	0.29	0.16
	200,000	1533	16.12	−2.87	0.36	0.21	0.34	0.19
	400,000	1019	10.71	−3.58	0.24	0.13	0.47	0.27
DSCC	63.5%	3330	35.02	−3.24	0.46	0.26	0.25	0.14
	73.5%	1640	17.19	−3.71	0.26	0.15	0.44	0.25
PLCC	66,000	2168	22.80	−2.67	0.44	0.25	0.27	0.15
	108,000	1765	18.14	−2.91	0.37	0.21	0.33	0.19

## Data Availability

Not applicable.

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
