# Peer review of "Total and Differential Cell Counts as a Tool to Identify Intramammary Infections in Cows after Calving"

_animals, 2021, doi:10.3390/ani11030727_

Round 1

Reviewer 1 Report

The authors assessed the predictive values of SCC and DSCC (as well as, PLCC) and their ability to detect intramammary infections in dairy cows, as well as their potential economic impact. Although, the authors deal with a large database and deal with an important issue for dairy chain, there are many concerns that should be addressed:

Major revisions:

1) The bacteriological analysis was carried out just using 18 h incubation. Why? Furthermore, the authors classified the results based on the mastitis pathogens isolated and the number of colonies retrieved on bacteriological examination. Why the authors used these criteria? (e.g., just consider bacteriological positive when > 5 colonies of environmental streptococci or Gram-negative bacteria were isolated). Moreover, it could be hard to say that all other environmental streptococci are minor pathogens (please, see: Rodrigues et al., Journal of Dairy Science, 2016; Scillieri Smith et al., Journal of Dairy Science, 2020). Please, include (a) proper reference(s) for the criteria used. In the opinion of this reviewer, it could be a limitation of this study, and may impact and explain, at least in part, the results obtained;

2) It the opinion of this reviewer, it is not reasonable to deal with all samples (< 30 days in lactation) as a homogenous sampling (e.g., colostrum, transition milk and milk together). For instance, the dynamic of milk SCC in the first days after calving is so high. For instance, the inclusion of colostrum and transition milk (high SCC even in healthy quarters) together with milk samples could contribute to the high somatic cell count threshold found. Please, try to better discriminate the quarter milk samples used;

2) Please, describe the possible limitations of method used (the non-discrimination of lymphocytes and PMNs);

3) Please, consider including some important studies (e.g., Line 71) that deal with colostrum/milk samples after calving (e.g., Ferronatto et al., Italian Journal of Animal Science, 2018; Lozato et al., Journal of Dairy Science, 2020). Please, also consider discussing our data with those studies;

4) I also recommend the authors to perform the statistical analysis including quarter milk samples divided just by their bacteriological outcomes (e.g., overall positive and negative outcomes);

5) Regarding the results of the present study, the DSCC and PLCC did not improve the diagnostic performance of intramammary infections at quarter level. Maybe, the redundancy of the immune response may explain these outcomes. Please, discuss about it (please see: Leitner et al., PloS One, 2015);

6) Please, discuss about the higher SCC threshold found for primiparous cows (e.g., De Vliegher et al., Journal of Dairy Science, 2012; Zecconi et al., Italian Journal of Animal Sciences, 2018; Persson Waller et al., Journal of Dairy Science, 2020);

7) Did the authors consider creating an index by dividing the percentage of PMN + Lymphocytes by the percentage of Macrophages, as well as their PLCC (percentage x total number).

Minor revisions:

1) Keywords: I suggest the authors to not include selective dry cow therapy, as its effect was not evaluated (the study was not designed to evaluate any effect of selective dry cow therapy, no data before drying-off was showed). Maybe, periparturient period or colostrum, may be (a) better keyword(s);

2) Please, change “ml” to “mL”;

3) Line 51: In the opinion of this reviewer, it is not suitable to say the term “udder inflammation”, as bacteria could be isolated in aseptically collected milk samples from healthy quarters (e.g., non-aureus staphylococci, please see: Traversari et al., Frontiers in Veterinary Science, 2019; Wald et al., Animals, 2019; Porcellato et al., Scientific Reports, 2020);

4) Table 2: ‘Columns” or ‘Lines”?

Reviewer 2 Report

In this report, Zecconi et al. report a new marker to identify cows at higher risk, post-calving, of being infected by major mastitis pathogens. This marker is the product of SCC by the percentage of (PMN+LYM) given by the DC7 milk analyzer.

The predictive value of this new marker to identify MP infected cows is compared to that of the SCC and DSCC markers. The cost benefit of applying the new PLCC marker to select cows for bacteriological analysis is analyzed.

L65-72: This paragraph could be greatly improved. The objective of the present report should be clarified. Per se SDCT does not require control of udder health post-calving. Control of udder health post-calving is an additional measure to be taken in order to improve animal health at the herd level. One way to do so is, as the authors state, to perform microbiological analyses but this is not feasible from a financial point of view since it would endure important additional costs. Therefore, breeders have to rely on tools such as SCC, CMT,… in order to do the microbiological analyses on animals with a high risk of being infected. This is where this report is interesting.

The legend for figure 1 needs to be clarified and a description of the different numbers should be given. Do the numbers indicate number of cases or cost or anything else ? Without such an explanation, the reader is unable to clearly evaluate the relevance of the scenario discussed.

Abbreviations used in Table 5 should also be clarified: what is the meaning of “P among test pos”, “Cont.”, “N. test”? The column “%” is percent of what ?

It is not clear in the description of the analyses of preventive costs if the authors have included the cost of MP mastitis that are not detected (false negative) using PLCC compared to SCC ? From Tables 3 and 4, FN are always lower for SCC than for PLCC which could translate into higher costs due to undetected MP infected animals when PLCC is used.

The sensitivity analysis should be performed for the two other markers (SCC and DSCC) in order to clearly demonstrate the benefits of the PLCC marker in different situations.

Minor comment:

How is the optimal value defined? For example, is It defined as the value for which the absolute value of the difference between the sensitivity and specificity values is minimum ? or for which the accuracy is maximum ?

Because E. coli is a coliform, the authors should use “Other coliforms” in their different tables when alluding to Gram negative strains other than E. coli.

How do the authors explain the fact that PLCC is higher than SCC for S. uberis since PLCC, in all cases, should be lower than SCC ?

The manuscript should be thoroughly checked for grammatical errors. Here are a few examples:

L49: “and to the proportion of cows”: should it be “and also the proportion of cows”

L175: “due to by” please rephrase

L240 : “did not show”

Other changes needed:

L50 and 53: 200.000 should be replaced by 200,000 and 100.000 by 100,000

L63: please clearly state the meaning of DHI : dairy herd improvement

Round 2

Reviewer 2 Report

The authors have addressed my concerns appropriately. Only a few typos remain: (in the abstract: 5-30 days; in the legend for figure 1 : red or blue).